# Two-dimensional topological superconductivity in Pb/Co/Si(111)

Gerbold C. Ménard[1,2], Sébastien Guissart[3], Christophe Brun[1], Raphaël T. Leriche[1], Mircea Trif[3], François Debontridder[1], Dominique Demaille[1], Dimitri Roditchev[1,4], Pascal Simon [3] & Tristan Cren[1]

Just like insulators can present topological phases characterized by Dirac edge states, superconductors can exhibit topological phases characterized by Majorana edge states. In particular, one-dimensional topological superconductors are predicted to host zero-energy Majorana fermions at their extremities. By contrast, two-dimensional superconductors have a one-dimensional boundary which would naturally lead to propagating Majorana edge states characterized by a Dirac-like dispersion. In this paper we present evidences of one-dimensional dispersive in-gap edge states surrounding a two-dimensional topological superconducting domain consisting of a monolayer of Pb covering magnetic Co–Si islands grown on Si(111). We interpret the measured dispersive in-gap states as a spatial topological transition with a gap closure. Our method could in principle be generalized to a large variety of heterostructures combining a Rashba superconductor with a magnetic layer in order to be used as a platform for engineering topological quantum phases.

[1] Institut des Nanosciences de Paris, Université Pierre et Marie Curie (UPMC) CNRS-UMR 7588, 4 Place Jussieu, 75252 Paris, France. [2] Center for Quantum Devices and Station Q Copenhagen, Niels Bohr Institute, University of Copenhagen, Universitetsparken 5, 2100 Copenhagen, Denmark. [3] Laboratoire de Physique des Solides, CNRS, Univ. Paris-Sud, Université Paris-Saclay, 91405 Orsay Cedex, France. [4] Laboratoire de physique et d'étude des matériaux, LPEM-UMR8213/CNRS-ESPCI ParisTech-UPMC, 10 rue Vauquelin, 75005 Paris, France. Correspondence and requests for materials should be addressed to P.S. (email: pascal.simon@u-psud.fr) or to T.C. (email: tristan.cren@upmc.fr)

The examination of supposedly well-understood condensed matter systems through the prism of topology has led to the discovery of new quantum phenomena that were previously overlooked. Just like insulators can present topological phases characterized by Dirac edge states, superconductors can exhibit topological phases characterized by Majorana edge states. In particular, one-dimensional topological superconductors are predicted to host zero-energy Majorana fermions at their extremities. Zero-bias anomalies localized at the edge of proximity induced superconducting wires were recently interpreted as fingerprints of the emergence of topological superconductivity[1–3]. By contrast, two-dimensional (2D) superconductors have a one-dimensional boundary which would naturally lead to propagating one-dimensional Majorana edge states characterized by a Dirac-like dispersion.

It has been realized during the past decade that our understanding of insulators and superconductors should be revised according to the topological properties of these materials. Superconductors with a fully opened gap can indeed be classified by some integer index[4–6]. Among the different types of superconductors, 2D time-reversal invariant topological superconductors are characterized by a $\mathbb{Z}_2$ invariant (0 or 1) that distinguishes between the trivial and topological phases[7, 8]. When time-reversal symmetry (TRS) is broken, several topological phases are possible and can be labeled by an integer number $Z$ that corresponds to the number of Majorana chiral edge modes in the system[7, 8].

Among the various known 2D superconductors, Pb monolayers grown on semiconducting substrates appear as ideal candidates to induce 2D topological superconductivity[9]. Indeed they are are known to present a strong Rashba spin–orbit coupling[10–13] that should induce a hybrid singlet-triplet order parameter[14, 15] in the superconducting state. One direct manifestation of the hybrid pairing induced by the spin-orbit interaction is the huge increase of the in-plane critical fields observed in monolayer of Pb/GaAs(110)[10]. Such a singlet–triplet hybrid system would be a topological superconductor if the triplet component was stronger than the singlet one. However, the absence of in-gap edge states at the interface between the Pb/Si (111) monolayer and trivial Pb islands implies that this system is not topological on its own[16]. An additional ingredient is therefore required for this system to enter a topological regime. Local magnetism is a promising route to achieve this goal.

In superconductors, point-like magnetic impurities give rise to Shiba bound states[17, 18] which have been shown to possess a long-range spatial extent in 2D[19, 20]. Arrays of magnetic impurities are however expected to give rise to chiral topological superconductivity[21–23], and the presence of a strong spin–orbit coupling may actually enhance this tendency[24, 25]. Recently, Nadj-Perge et al.[2] observed zero-bias anomalies strongly localized around one edge of chains of magnetic Fe atoms deposited on top of bulk Pb (110). These spectroscopic signatures[2, 26, 27] might be fingerprints of Majorana bound states equivalent to those originally observed by Mourik et al.[1] in proximity induced superconducting Al-InAs nanowires under parallel magnetic field.

Here, in this experimental and theoretical work, we give evidence that the strong local exchange field created by a

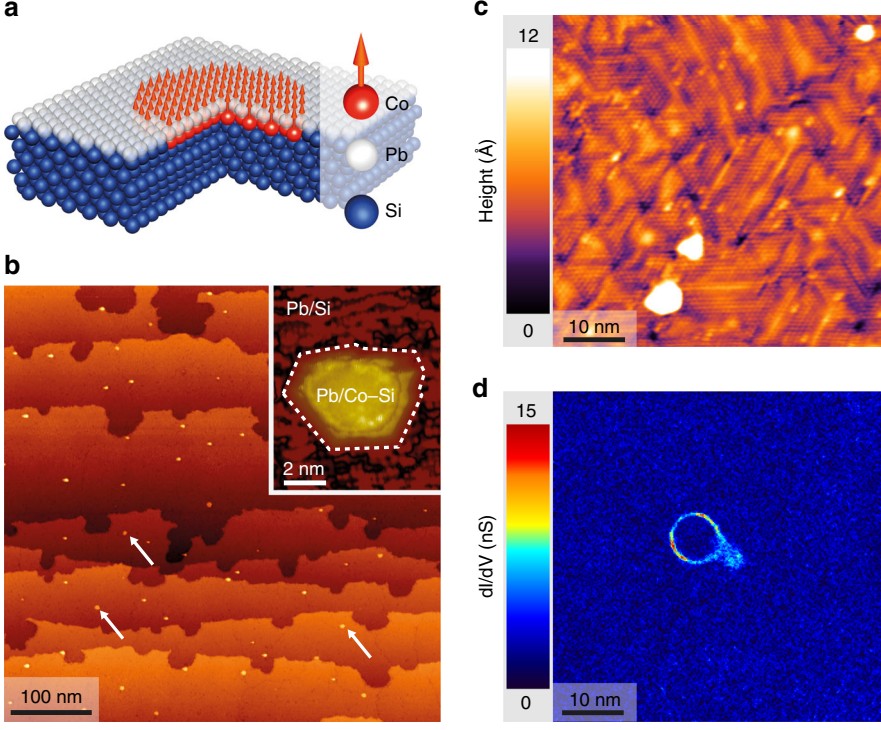

**Fig. 1** Superconducting-ferromagnetic hybrid system Pb/Co/Si(111). **a** Schematic structure of the system investigated by STM. A monolayer of Pb is grown on top of Si(111) at the surface of which magnetic disks of Co–Si have grown. **b** Large scale topography of the sample after a long annealing at 400 °C which provokes the formation of a Pb defected mosaic phase everywhere and reveal the buried Co–Si clusters. The magnetic clusters are revealed as small dots of 5–10 nm width. Inset: zoom on such a revealed Co–Si cluster. A highly inhomogeneous and defective Pb/Si(111) mosaic phase is seen outside the island. **c** Topography of a $63 \times 63$ nm$^2$ area measured by scanning tunneling microscopy (bias voltage −50 mV, tunnel current 30 pA). The atomic pattern and corrugation reveals a striped-incommensurate Pb/Si(111) monolayer. A buried Co–Si cluster is present in this area below the monolayer but it does not appear in the topography. **d** d$I$/d$V(V)$ conductance map at 1.32 meV measured at 300 mK on the area shown in image **c**. The experiment was performed with a superconducting Pb tip, thus the energy is shifted by the gap of the tip. Once deconvoluted from the tip density-of-states, the map displays ring-shaped in-gap states located at the Fermi level (see text and Supplementary Fig. S1 for details). Thus, the ring-like feature corresponds to a gapless region but everywhere else a hard superconducting gap is present as evidenced by the homogeneous dark-blue color (low conductance)

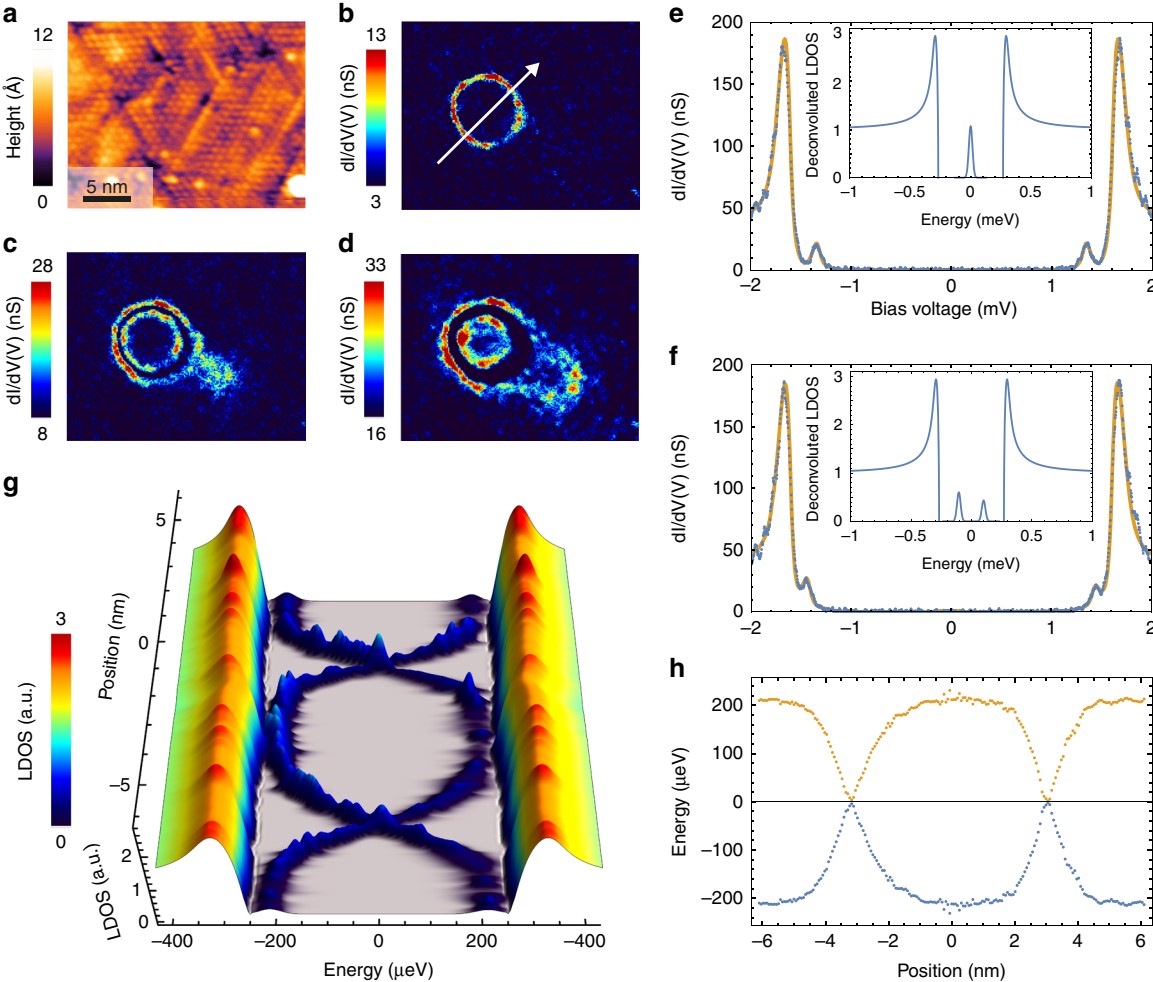

**Fig. 2** Topological edge states. **a** Topography of an area $23 \times 18$ nm$^2$ measured by scanning tunneling microscopy (bias voltage −50 mV, tunnel current 30 pA). **b**–**d** Scanning tunneling spectroscopy d$I$/d$V(V)$ conductance maps of the same area at three different voltage biases (1.30 meV, 1.43 meV, 1.5 meV, respectively) showing the energy evolution of the edge states appearing at the frontier between a topological and a trivial superconductor. This area corresponds to the same region of the sample as the one shown on image **a**. **e** The blue dotted curve shows a d$I$/d$V(V)$ conductance spectrum measured at the intersection between the cut and the ring shown in **b**. The conductance spectra are a convolution of the sample LDOS with the BCS gap of the Pb superconducting tip. The orange curve is a fit of this convoluted LDOS. The corresponding deconvoluted LDOS is shown in the inset. It exhibits a pic at the Fermi level. **f** The same as **e** but for a spectrum measured in the upper part of the outer ring shown in image **c**. The deconvoluted LDOS shown in the inset displays two in-gap peaks at symmetric energies. **g** Line-cut of the deconvoluted LDOS along the dotted line in **b** showing the spatial dispersion of the topological edge states. The edge states displays a X-shape at the interface of the cluster. **h** Energy position of the in-gap states extracted from the fit of the linecut shown in **g**, no anti-crossing is observed

single magnetic island locally triggers a transition between trivial and topological superconductivity in a two-dimensional Rashba system. Using scanning tunneling spectroscopy at 300 mK, we present evidence of 1D dispersive in-gap edge states surrounding 2D topological superconducting domains. The topological domains consists of a monolayer of Pb covering magnetic Co–Si islands grown on Si(111). We interpret the measured dispersive in-gap states as a spatial topological transition with a gap closure. Our method could in principle be generalized to a large variety of heterostructures combining a Rashba superconductor with a magnetic layer in order to be used as a platform for engineering topological quantum phases.

## Results

**Sample preparation.** We look for a topological transition by studying magnetic domains embedded in a Pb monolayer (see Fig. 1a). The samples studied here were grown by sequential

evaporation of Co and Pb on Si(111) at room temperature followed by an annealing at 375 °C (see Methods). During the annealing, the Co atoms diffuse into the first Si layers and form Co–Si clusters. DFT calculations indicate that Co-Si clusters are magnetic for any Co concentration[28]. This magnetization is confirmed by magneto-optical Kerr effect measurements of annealed Co/Si(111) ultrathin films[29, 30]. Our preparation results in a Pb monolayer covering subsurface Co–Si magnetic domains having diameters ranging between 5 and 10 nm randomly distributed all over the sample, as seen in bright-field transmission electron microscopy (see Supplementary Note 6). The Pb monolayer is prepared in the striped incommensurate phase[9, 31], has a nominal coverage of 1.30 monolayer and a critical temperature of 1.6–1.8 K for a gap $\Delta = 0.290 - 0.350$ meV[11, 32], depending on the disorder and precise Pb coverage. The Co–Si domains do not manifest directly in the STM topography where only the top Pb layer reconstructed into the striped incommensurate phase is visible (see Fig. 1c). However, after a long annealing at 400 °C, the Co–Si clusters are revealed after most of

the Pb has evaporated (see Fig. 1b and more details in Supplementary Note 6).

**1D dispersive in-gap edge states measured by STS.** In order to determine how the superconducting order is affected by the Co–Si domains we performed scanning tunneling spectroscopy experiments at 300 mK with a Pb superconducting tip, which enables a high energy resolution. On large scale tunneling conductance maps, we observe sharp structures that appear more or less ring-shaped (see Fig. 1d and Supplementary Fig. S5). The size and the shape of the spectral features are consistent with the Co–Si clusters seen in TEM on as-grown samples and by STM on long-annealed samples. Thus we estimate that the shape of the spectral features is given by the geometry of the generating Co–Si cluster. Since most of the clusters are disk-shaped the features in the LDOS appear as rings. For simplicity we will label these features as "rings" throughout the manuscript, although our argumentation also holds for less isotropic geometries. A zoom of the ring-shaped structure is presented in Fig. 2a–d. The ring shown in Fig. 2b appears in conductance maps taken around $V = 1.32$ mV instead of $V = 0$. Nevertheless, it actually corresponds to a structure in the local density of states (LDOS) of the sample close to $E_F$. This difference between the applied bias ($V = 1.32$ mV) and the actual energy ($V = 0$ meV) of a feature inside the superconducting gap is due to the fact that the tunnel conductance is a convolution of the LDOS of the sample with the BCS density of state of the superconducting tip. To assert this, a spectrum measured locally on the ring is shown in Fig. 2e. This conductance curve was fitted by taking into account the tip gap which was measured at 2.05 K, i.e., above the critical temperature of the Pb monolayer (see Supplementary Note 1 and 2). The deconvoluted LDOS of a spectrum taken on the ring is shown in the inset of Fig. 2e. There, a peak localized at $E_F$ with a half-width-at-half-maximum of the order of 20 µeV is clearly seen. The ring-shaped spectroscopic features appearing at the Fermi energy can also be measured using a normal Pt tip at 300 mK. In that case the energy resolution 90 µeV is not as good as the one with a Pb tip (see Supplementary Note 8). The spectra measured on the ring with a normal tip show a peak localized exactly at zero energy, in the middle of the gap, in perfect agreement with the LDOS obtained by deconvolution.

The thickness of the ring seen at $E_F$ is approximately 0.7 nm which is comparable to the typical atomic dimensions and Fermi wavelength of the system[33]. This spatial extent is much smaller than the coherence length $\xi$ ($\simeq 50$ nm), the mean free path $\ell$ ($\simeq 4$ nm) of Pb/Si(111)[11] and the size $R_c$ ($\simeq 5$–10 nm) of the Co–Si cluster. This is remarkable since for superconductors in the diffusive limit $\ell \ll \xi$, the typical length scale of superconducting variations is given by the coherence length $\xi$. At finite energy inside the deconvoluted gap, the sharp line seen in the conductance map at $E_F$ splits into two concentric features, one moving inward, the other moving outward (see Fig. 2b–d). A spectrum taken on the inner structure of Fig. 2c is shown in Fig. 2f. The deconvoluted LDOS shows two peaks in the gap at $\pm 105$ µeV with a slightly different amplitude. Another important feature observed at $|E| \lesssim \Delta$ is that the sensitivity to disorder increases as the bias voltage is moved towards the gap energy and the edge states get closer to the bulk states (Fig. 2d).

To get a better grasp of the way these states disperse in space and energy we present on Fig. 2g, h a line-cut of the $dI/dV$ maps passing through the center of the domain as a function of energy, deconvoluted from the superconducting tip (see Supplementary Note 1). A similar dispersion measured with a normal tip is shown in Supplementary Fig. S8. One important fingerprint highlighted in the profiles of Fig. 2g, h is the crossing of the states

connecting both sides of the superconducting gap. The X-shape of the dispersion in real-space clearly shows that there is no measurable anti-crossing at the Fermi energy.

**Discarding non topological scenarios.** Among the possible explanations of the spectroscopic features that we observe around Co–Si domains, the most simple would be to consider them as Yu–shiba–Rusinov (YSR) bound states induced by magnetic impurities. This cannot be the case as we explain now. YSR bound states, that are routinely observed around individual magnetic impurities in SIC–Pb/Si(111) have some very different signatures. Spectroscopic maps of YSR bound states show an irregular speckle like pattern which extends up to several tens of nanometers from the magnetic impurity and decays like $1/\sqrt{r} \exp(-r/\xi)$[19] (see Supplementary Note 7). The spectra taken on top of an atomic impurity or at several tens of nanometers from it show in gap states exactly at the same energy, no energy dispersion is observed. Thus the strongly dispersive character of the in-gap states observed around Co–Si clusters cannot be attributed to usual YSR bound states since the latter are non-dispersive.

The X shape crossing of the dispersive states in the gap presents analogies with what was observed in superconducting-normal-superconducting (S–N–S) devices by Pillet et al.[34] where Andreev states were induced in the N part. The energy of the Andreev bound states was changed and some crossing of the Fermi energy were observed as function of a gate voltage. If the Co–Si clusters would locally create a normal area in the topmost Pb layer, one could imagine that Andreev states would be induced in the N domain and that the STM tip could act as a local gate. However, the Pb layer is in diffusive limit ($\xi \gg \ell$, where $\xi \approx 50$ nm is the coherence length and $\ell \approx 4$ nm the mean free path). Therefore, any proximity effect would lead to the appearance of continuous LDOS with a minigap in the N domain instead of a few discrete Andreev bound states[35]. Moreover, contrary to what we measure, Andreev states should be localized only in the N domain and should not extend far away in the surrounding superconducting area as we observe for the dispersive states around Co–Si domains.

**Strong arguments supporting 2D topological superconductivity.** The simplest interpretation of our experimental findings is to consider that the magnetic Co–Si clusters drives the Pb/Si(111) monolayer into a topological superconducting state. This simple assumption could explain why the in-gap states shown in Fig. 1d are prevented from propagating inside or outside the ring in the same way surface states are confined in the case of topological insulators. The observed ring-like features would be related to edge states surrounding a topological domain (note that their spatial width is much smaller than the cluster size, thus preventing any overlap between states of opposite side of the cluster). These states would appear as due to the topological transition triggered by the underlying magnetic cluster that defines two areas, a topological one above the cluster and a trivial one elsewhere. The absence of anti-crossing further calls for the topological nature of these edge states. If we were dealing with trivial in-gap states in a Zeeman field, the strong disorder of the incommensurate Pb/Si (111) samples (see Fig. 1c) would indeed lead to an anti-crossing. However, chiral topological dispersive states, which are protected against disorder, would not display any[7, 8].

The fact that the striped incommensurate Pb/Si(111) phase is a trivial superconductor possessing a strong Rashba spin orbit-coupling[12, 13] is a strong hint for a topological transition under the influence of a local exchange field induced by an underlying insulating magnetic cluster. According to the most

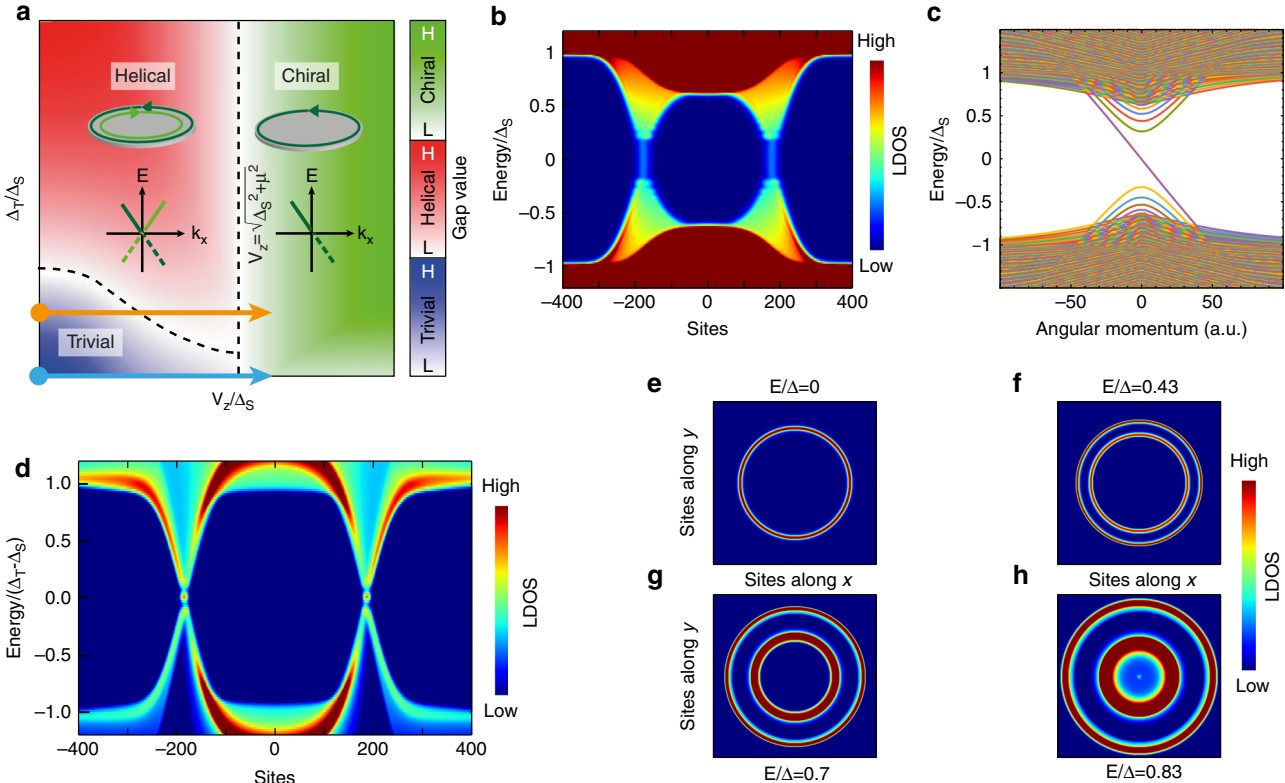

**Fig. 3** Theoretical calculations. **a** Phase diagram of 2D topological superconductivity as a function of the Zeeman field $V_z$ and the triplet order parameter amplitude $\Delta_T$. The black dashed lines show the transitions between the trivial and helical phases, as well as the transition between the helical and chiral phases. The insets show the typical behavior of the edge states for the helical case (with two counter-propagating edge states) and the chiral case (only one edge state whose chirality is determined by the orientation of the magnetic field). The color code corresponds to the values of the gap in each phase with one color for each phase. **b** Real space tight binding calculation of $\rho(r, E)$ showing a Majorana edge state dispersing throughout the gap (parameters used for the calculation: $t = 100$, $\mu = 50$, $\alpha = 200$ and $\Delta_S = 140$). No triplet pairing was used: $\Delta_T = 0$. The diameter of the system is $D = 500$ sites. The Zeeman potential is given by $V_z(r) = 120(1 - \tanh(r - R_c)/w)$, with $R_c = D/2$ and $w = 0.14D$. **c** Energy dispersion as function of the angular momentum $E(k_\theta)$ with the same parameters used for the tight binding calculation in **b** showing a Majorana branch closing the gap. **d** Real space tight binding calculation of $\rho(r, E)$ in the presence of a triplet pairing (Parameters used for the calculation: $t = 100$, $\mu = 50$, $\Delta_S = 150$, $\Delta_T = 60$ and $\alpha = 0$). The Zeeman potential is the same as in **b**. Note that in all our numerical calculations, the X-shape features originate from a real space partial gap closing. This can be traced back to the smoothness of the Zeeman field decaying over a scale $w \gg k_F^{-1}$. In the more realistic limit, $w \ll R_c < \xi$, we expect that only the edges states will cross the gap and give rise to these X-shape features. **e–h** Calculated LDOS with the same parameters than in **d** for 4 different energies: $E/\Delta = 0$, 0.43, 0.7, and 0.83

recent DFT calculations[12] for the commensurate $\sqrt{3} \times \sqrt{3}$ monolayer of Pb/Si(111), which is the closest crystalline structure that approximates the sample we study, the band structure can be very well approximated by a single parabolic band with the spin degeneracy lifted by a strong Rashba spin orbit coupling. The band structure of the striped-incommensurate phase was recently measured by spin-resolved ARPES[13] and a Rashba-like band structure was indeed observed around the center of the Brillouin zone Γ. In addition, some tiny spin-splitted bands where also observed close to the $K$ and $K'$ points. These four tiny bands (two at $K$ and two at $K'$) cannot change the parity of the number of Fermi sheets at the Fermi level under the effect of a Zeeman splitting, thus they should not play any role in the topological transition. By contrast the dominating Rashba bands at Γ can change the parity of the number of Fermi sheets if the Zeeman field is large enough in order to shift one of the two Rashba bands above the Fermi level. These minimal ingredients combining Zeeman field with Rashba spin–orbit coupling can explain the appearance of chiral superconductivity above Co–Si domains, as we develop below in a theoretical model.

**Modeling 2D topological superconductivity.** We consider the following Hamiltonian in which we include the minimal

ingredients to explain our experimental observations:

$$
\begin{aligned}
H = \; & \left[ (p^2/2m - \mu) + \alpha(\boldsymbol{p} \times \boldsymbol{\sigma})_z \right] \tau_z + V_z(\boldsymbol{r}) \sigma_z \\
& + \left[ \Delta_S + (\Delta_T/k_F)(\boldsymbol{p} \times \boldsymbol{\sigma})_z \right] \tau_x.
\end{aligned} \quad (1)
$$

The first term stands for the kinetic energy, with $m$ the effective electron mass and $\mu$ the chemical potential. The second term is the Rashba spin-orbit coupling of strength $\alpha$. The third term reflects the magnetic exchange coupling caused by the Co–Si cluster which we describe as a spatially varying Zeeman term $V_z(r)$, while the last two terms describe the s- and p-wave pairings, of strengths $\Delta_S$ and $\Delta_T$ respectively (see Supplementary Note 3 for details). Here $\boldsymbol{\sigma} = (\sigma_x, \sigma_y, \sigma_z)$ and $\boldsymbol{\tau} = (\tau_x, \tau_y, \tau_z)$ are Pauli matrices that act in the spin and Nambu (particle-hole) spaces respectively. The Nambu basis spinor is given by $\boldsymbol{\Psi}_p^\dagger = \left( \hat{\psi}_{p\uparrow}^\dagger, \hat{\psi}_{p\downarrow}^\dagger, \hat{\psi}_{-p\downarrow}, -\hat{\psi}_{-p\uparrow} \right)$. Edelstein[14], followed by Gorkov and Rashba[15] showed that, in the presence of a Rashba spin–orbit coupling, the electron–phonon interaction responsible for the formation of Cooper pairs should lead to a mixed singlet–triplet order parameter. A spin triplet pairing term $\Delta_T$, aligned with the Rashba vector, has thus been added to the Hamiltonian.

In order to characterize the different phases of a system described by the Hamiltonian given in Eq. (1), we first consider a

constant out-of plane Zeeman field $V_z(\mathbf{r}) \equiv V_z$. The phase diagram of this model is displayed in Fig. 3a. Three different phases appear as function of the amplitude of the Zeeman splitting and of the triplet pairing. The two dashed lines correspond to transitions where the superconducting gap closes and a Dirac-like conic dispersion appears around the Fermi level[36].

For $\Delta_T = 0$, the system undergoes a first transition from a trivial state to a topological chiral state at the Zeeman field $V_z = \sqrt{\Delta_S^2 + \mu^2}$[37]. Below this threshold the superconducting state is trivial, with a $C = 0$ Chern number. Above we reach the domain of chiral topological superconductivity with a Chern number $C = +1$. Contrary to the 1D case, where Majorana bound states appear at the extremities of a topological wire, for a 2D system, a Majorana chiral dispersion will appear at the edge of the topological domain (see inset of Fig. 3a).

In order to simulate our experimental observations in real space, we need to consider a spatially varying magnetic field $V_z(r)$. We model the cluster as a strong Zeeman field which locally drives the system into a topological chiral superconducting phase. We consider a cylindrical symmetry for the system (see Supplementary Note 5) with a radial Zeeman field $V_z(r)$. We take a profile for $V_z(r)$ such as to have a Zeeman field varying from $V_z = V_{zmax} > \sqrt{\Delta_S^2 + \mu^2}$ on top of the cluster to $V_z = 0$ away from the cluster. This realizes a continuous path in the phase diagram drawn in Fig. 3a, which goes from a chiral superconducting phase to a trivial one (blue arrow). In Fig. 3c, we show the dispersion relation with the momentum $k_\theta$ (conjugate to the perimeter of the magnetic disk) obtained by diagonalizing Eq. (1) with a gaussian Zeeman profile. A chiral Majorana edge state crossing the gap is clearly identified. We also computed the local density of states $\rho(E, r)$ at energy $E$ and position $r$ so as to compare with the data measured by STM. The LDOS profile shown in Fig. 3b corresponds to a linecut through a magnetic cluster with a topological chiral state on the inside and a trivial state everywhere else. The two phases with non equivalent topological orders are spatially separated by a Majorana branch (appearing as a straight vertical line) that disperses throughout the gap. As can be seen, the agreement with the data shown in Fig. 2g is perfectible. The addition of a small triplet pairing component improves the agreement as we show now.

The previous calculations assumed $\Delta_T = 0$. A finite triplet pairing $\Delta_T$ produces a richer phase diagram shown in Fig. 3a. In particular, for $V_z = 0$, the hamiltonian respects time-reversal symmetry (TRS) and a topological transition between a trivial state and a helical state appears at $\Delta_T > \Delta_S$. Due to the different topological indexes between the topological superconductor and the vacuum, a Kramers pair of helical edge states appears at the edge of the topological domain, as shown in the inset of Fig. 3a: these states are dispersive Majorana fermions[7, 8]. Applying a magnetic field $V_z$ breaks TRS but does not destroy the helical behavior of the edges. The magnetic field only makes these edges inequivalent, or "quasi-helical", so that they are propagating with different velocities and can separate spatially. A second topological transition occurs at $V_z = \sqrt{\Delta_S^2 + \mu^2}$ independently of $\Delta_T$[38], after which the system becomes purely chiral. In this phase, only one propagating Majorana edge state exists, exactly as in the previous case where no triplet pairing was present. This is at odds with the helical case that exhibits a pair of states with opposite chiralities. Therefore, chiral superconductivity compares to quantum Hall effect in the same way helical superconductivity compares to quantum spin Hall effect[7]. We should stress that the "quasi-helical" phase at $V_z \neq 0$ is only weakly topologically protected due to TRS breaking (see Supplementary Note 4).

## Discussion

Experimentally, we found no evidence of topological super-conductivity far away from the magnetic cluster (no in-gap edge modes were found at the interface between the Pb/Si(111) monolayer and bulk-like Pb islands[16]). This implies $\Delta_T < \Delta_S$ for our model. Moreover, the superconducting gap measured in the Pb/Si(111) monolayer, far away from the magnetic cluster, $\Delta \sim 0.3$ meV should correspond to the effective gap $\Delta = |\Delta_S - \Delta_T|$. We find a good qualitative agreement with our experimental data by assuming that the cluster generates a strong Zeeman field which locally drives the system from a chiral topological superconducting phase to the trivial one passing via some intermediate (quasi-)helical phase (path shown in Fig. 3a with an orange arrow). Our numerical calculations of $\rho(E, r)$ are shown in Fig. 3d–h. It qualitatively reproduces the X shaped real-space crossing displayed in Fig. 2g, h. It appears within this set of parameters as a partial gap closure with associated dispersive edge states, the chiral Majorana mode being the only robust and topologically protected one. Very recent calculations, done in our geometry with a BdG model on a lattice incorporating a Rashba coupling or a triplet pairing, confirm the formation of a X-shape in the dispersion with the appearance of a single ring at the Fermi energy that splits into two rings when the energy is increased[39]. We expect that more realistic calculations performed with a range of parameters such that $R_c < \xi$ instead of $\xi \ll R_c$, $R_c$ being the size of the cluster and $\xi$ the coherence length, would improve the agreement with the data by giving a hard gap that remains finite while a chiral Majorana band crosses the gap possibly accompanied by quasi-helical states.

In conclusion, by using a 2D Rashba superconductor coupled to a magnetic cluster we observed the realization of 2D topological superconductivity through the appearance of dispersive in-gap edge states surrounding the cluster. The model we developed qualitatively reproduces the experimental results and does not require any fine tuning of the parameters. The simplicity and generality of this model proves that it should be easily generalized to many other systems. The next step would be to observe Majorana bound states in vortex cores of the 2D topological domains. Such discovery could be be of great interest in the current development of quantum electronics based on the braiding of Majorana bound states.

## Methods

**Sample preparation.** The $7 \times 7$ reconstructed $n$-Si(111) (room temperature resistivity of few m$\Omega$ cm) was prepared by direct current heating to 1200 °C followed by an annealing procedure driving the temperature from 900 to 500 °C. Subsequently, $10^{-3}$ monolayers of Co were evaporated in 6 s on the $7 \times 7$ reconstructed Si(111) substrate kept at room temperature. The Co was evaporated from an electron beam evaporator calibrated with a quartz micro-balance. Four monolayers of Pb were then evaporated using another electron beam evaporator. The Pb overlayer was formed by annealing the sample at 375 °C for 90 s by direct current heating. This step leads to a striped incommensurate (SIC) reconstruction of the Pb monolayer. At no stage of the sample preparation did the pressure exceed $P = 3 \times 10^{-10}$ mbar.

**Measurements.** The scanning tunneling spectroscopy measurements were performed in-situ using a home-made apparatus at a base temperature of 280 mK and in ultrahigh vacuum in the low $10^{-11}$ mbar range. Mechanically sharpened Pt/Ir tips were used. These tips were made superconducting by crashing them into silicon carbides protrusions covered with Pb. The resulting superconducting gap of the tip was measured and extracted from both S–S and S–N tunneling spectra at 280 mK and 2.05 K, respectively, when the SIC Pb overlayer is in its superconducting and normal state. The bias voltage was applied to the sample with respect to the tip. Typical set-point parameters for topography are 20 pA at $V = -50$ mV. Typical set-point parameters for spectroscopy are 120 pA at $V = -5$ mV. The electron temperature was estimated to be 360 mK. The tunneling conductance curves d$I$/d$V$ were numerically differentiated from raw $I(V)$ experimental data. Each conductance map is extracted from a set of data consisting of spectroscopic $I(V)$ curves measured at each point of a $220 \times 220$ grid, acquired

simultaneously with the topographic image. Each $I(V)$ curve contains 2000 energy points in the [−5; +5] meV energy range.

**Data availability**. The data that support the findings of this study are available from the corresponding authors upon request.

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

## Acknowledgements
This work was supported by the French Agence Nationale de la Recherche through the contract ANR Mistral (ANR-14-CE32-0021). G.C.M. acknowledges funding from the CFM foundation providing his Ph.D. grant. We thank L.L. for complementary TEM measurements.

## Author contributions
G.C.M., C.B., R.T.L., and T.C. carried out the STM/STS experiments and D.D. performed the TEM measurements. G.C.M. and T.C. processed and analyzed the data. P.S., S.G., G.C.M., M.T., and T.C. performed the theoretical modeling. All authors discussed the results and participated in the writing of the manuscript.

## Additional information

**Competing interests:** The authors declare no competing financial interests.

