## [Peer Review File · Nature Communications]

Reviewer #1 (Remarks to the Author):

The authors have revised their manuscript and have addressed most of my criticism, but not always in a clear and efficient manner.

We are sorry, we acknowledge that our responses were too long and probably not focused enough.

Still, some of the formulations are not very clear or explicit and could be made more straight forward. For example to my comment 2 regarding the shape of the circular structures being given by the shape of the Co clusters, the authors give a very lengthy explanation why the clusters are round. This was not my question. My request was to write that the features in the LDOS appear as circles because the clusters are round objects. In the response the authors write that they added the formulation "...is consistent with the Co-Si clusters seen in TEM and STM on long annealed samples". First of all this quote is not correct. In the manuscript it reads "is consistent with the SIZE OF THE Co-Si clusters seen in TEM and STM on long annealed samples" which is already much better, but still misses the main point. My comment was about the ring shape that appeared in the example given as perfectly ring shaped. The numerous repetitions of the fact that they are ring shaped suggested (and still suggests) that this would be a fundamental aspect. But in fact it is not. It is a mere consequence of the geometry of the clusters. Ring shapes may have manifold reasons, e.g. they may reflect a property of the substrate or in momentum space (as the authors showed themselves in one of their earlier work) rather than by a geometrical effect of the nanoobject. So, this point must be made clear! Luckily the authors did not only write the sentence quoted above but also the preceding sentence in the manuscript: "On large scale tunneling conductance maps we observe sharp structures that appear more or less ring-shaped (...) depending on the exact structure of the underlying object". I would prefer a clearer formulation as e.g. the following: The shape of the spectral features is given by the geometry of the generating Co-Si cluster. Since most of the clusters are disk-shaped the features in the LDOS appear as rings. For simplicity we will label these features as 'rings' throughout the manuscript, although all argumentation also hold for less isotropic geometries.

We understand your point and we have reformulated this part in the way you propose.

I still find the similarity between the experimental Fig. 2 g and h and the theoretical Fig 3. b and d too suggestive. Furthermore the arrangement of the 8 panels with different aspect ratios in Fig. 3 is not appealing to the eye, but this is not crucial. And the labels in the insets in Fig.2 e-f- and the color bars in a-d are too small.....

We have reshaped this figure accordingly to your comment in the final version.

I am not sure that the responses to the other reviewers are all understandable and adequate. They are always very long and give a lot of new arguments that do not appear in the manuscript or the SI. I leave it to them to decide. In fact I am more sceptic regarding the presentation than before in view of the very lengthy explanations in the rebuttal. I feel that at least part of the responses is really necessary to understand the work and to justify the assumption and approaches. But if this is so, they should appear in the manuscript (or at least in the SI).

We agree with the Referee and added the new pieces of information regarding Co-Si growth, their measured and calculated magnetic properties. We have also added the calculated and measured band structure properties of the SIC-Pb/Si(111) in the main text. We also decided to add a full discussion of the different possible explanations of our experimental results in the main text, based on what we explained in our response. We finally discard these hypothetical explanations in favour of 2D topological superconductivity induced above the buried Co-Si disks.

It seems that the presentation of the data was not really mature when it was submitted initially. The responses also show that there are ongoing considerations regarding the adequate modeling of the data and one might consider waiting with the publication until this is all settled.

A correct model with the good order of magnitude of the coherence length will not be ready soon. All people trying to simulate the same kind of problem face the same limitations. In order to strengthen our interpretation we added a reference to the recent work by K. Bjornson and A. M. Black-Schaffer, arxiv:1709.09061, who performed some BdG simulation on a lattice for our problem. They found qualitatively the same kind of partial and energy dispersion for the edge states, with in particular a two ring structure that merges in a single ring at the Fermi energy. This new work confirms that our interpretation is probably the best one that can be offered now.

However, I think the topic is very relevant and the central message – the observation of Majorana modes at the edges of a 2D island - important and merits immediate publication in Nature Physics, although the presentation appears unnecessarily complicated at some point.

We thank you very much for your helpful comments that helped us to improve our article.

Reviewer #2 provided confidential remarks to the editors, raising continued concerns over whether these observations could come from non-topological bound states, as they did not think that the Majorana nature of the modes has been confirmed.

We added a discussion in the article in order to explain why our observations cannot be explained by a simple Andreev bound state mechanism assisted with a gate. We are surprised that the referee sticks to his first comment while we brought him some clear explanations why the mechanism he suggests is impossible in our system. Visibly the referee overestimates his knowledge on proximity effect. If he wants to refresh his understanding of proximity effect in a diffusive system we suggest him to read the very good chapter on this topic we published recently in the book: “The Oxford handbook of small superconductors”.

Reviewer #3 (Remarks to the Author):

In this revised manuscript and the rebuttal, the authors satisfactorily addressed some questions. However, some critical questions remain unanswered. Below I elaborate further.

(1) I am satisfied with the corrections and clarifications made in their description of deconvolution to extract inherent spectroscopic information of the sample. In fact I am very impressed with their

spectroscopic imaging result.

(2) The additional TEM result is nice. Nevertheless, no information on chemical analysis is provided which would be important to substantiate the claim for the existence of 2D Co islands below the single layer Pb film

As explained in our previous response we used one of the best scanning TEM available in France (TITAN Themis 200) in STEM-HAADF mode but we could not get chemical sensitivity due to the fact that the Co-Si clusters fade away under the highly convergent electron beam needed for such kind of analysis.

Meanwhile we did some additional STM measurements preparing CoSi₂ islands grown on Si(111) at 550°C. In a second step we have grown a SiC-Pb monolayer above CoSi₂ islands. Our new spectroscopy results confirm that the structure of the buried clusters presented in our paper is not CoSi₂, but most probably Co-Si, as inferred from the published studies on Co/Si(111) grown at intermediate temperature 300-400°C.

(3) Most critically, I believe the claim that the regions with monolayer Pb on top of the Co 2D islands (referred to as Pb-Co-Si 2D islands) are topological superconductors (TopoSC) cannot be substantiated at the current stage. Their idea for the formation Topo-SC is based on proximity coupling between superconductor and the magnetic domain.

This is wrong, please read the article carefully. We do not assume any proximity effect anywhere in our model. You probably mixed up with other articles that propose such a mechanism. Our Hamiltonian is explicitly given in the main text (see page 6) and supplementary material.

This general idea has been discussed in several theoretical papers including the one cited in Ref 21. But they provide no experimental support for this claim. For example, one has no idea about the strength and the direction of the Zeeman field (out of plane or in-plane).

As explained in the supplementary information, the direction of magnetization does not play any role in the second model we propose (no Rashba, but a combination of singlet and triplet pairing). We agree on the criticism regarding the unknown strength of the Zeeman field. However the Referee should note that his comment is also valid for all the published studies having reported hypothetical topological superconductivity in ferromagnetic nanowires deposited on top of a SC. Thus we estimate that your comment is not fair or should also apply to the recently published works in this area.

Moreover, another important condition is that there must be odd number of Rashba bands crossing the Fermi energy. Where is the experimental evidence for this?

We are very surprised by your comment. Indeed, we added several references that should completely fit your concerns about our choice to describe the band structure of the striped incommensurate phase of Pb/Si(111) with a simple Rashba band. We cited a paper of DFT simulations (Ren et al PRB 94, 075436, 2016) that perfectly supports our claims. We can understand that you prefer experimental proofs, thus we are pleased to announce that spin-ARPES measurements were done and published in PRB 96 035432 (2017), just after we sent our previous response. These measurements are a strong support to our model (4 additional spin splitted tiny

Fermi sheets at K and K' also appear, but they cannot change the parity of band crossing at the Fermi level, which is your main concern).

Note that all the papers published on nanowires assumed a band structure due to fully polarized suspended Fe chains without any consideration of the crystal field effect of the substrate. This is a rough approximation, but for a first trial it is ok. We estimate that in our case, we can also approximate the real band structure by a simpler model as a first trial. This is what we have done, based on the theoretical DFT paper of Ren et al PRB 94, 075436, 2016, assuming a simple Rashba band which was confirmed to be true by spin-ARPES in Brand et al. PRB 96 035432 (2017). In conclusion, here too, we think that the Referee is not fair with us.

In summary, the authors have presented a very nice spectroscopic study showing the existence of some kind of "edge mode" at the lateral interface between some structures. This edge mode may very well be the long-sought-after Majorana mode. But without first providing that topological superconductor exists in the regions of Pb covered hidden Co-Si 2D islands (discussed in my comment (3)) it would be premature to make such a claim.

We could understand that you are not 100% convinced, but we estimate that the level of proof that we provided is much superior to what was supplied with other recent paper in the field. We cannot solve all the problems we are facing in a first work, it will certainly take years before being able to have a microscopic understanding of all the aspects of the investigated system. Our general feeling after the two reports is that the Referee has decided to do some obstruction. Anyway we forgive the Referee; time will judge our work whatever Referee 3 wishes.